# A mechanosensory receptor required for food texture detection in *Drosophila*

Juan Antonio Sánchez-Alcañiz[1], Giovanna Zappia[1], Frédéric Marion-Poll[2,3] & Richard Benton[1]

Textural properties provide information on the ingestibility, digestibility and state of ripeness or decay of sources of nutrition. Compared with our understanding of the chemosensory assessment of food, little is known about the mechanisms of texture detection. Here we show that *Drosophila melanogaster* can discriminate food texture, avoiding substrates that are either too hard or too soft. Manipulations of food substrate properties and flies' chemosensory inputs indicate that texture preferences are revealed only in the presence of an appetitive stimulus, but are not because of changes in nutrient accessibility, suggesting that animals discriminate the substrates' mechanical characteristics. We show that texture preference requires NOMPC, a TRP-family mechanosensory channel. NOMPC localizes to the sensory dendrites of neurons housed within gustatory sensilla, and is essential for their mechanosensory-evoked responses. Our results identify a sensory pathway for texture detection and reveal the behavioural integration of chemical and physical qualities of food.

[1] Center for Integrative Genomics, Faculty of Biology and Medicine, Génopode Building, University of Lausanne, CH-1015 Lausanne, Switzerland. [2] UMR Evolution, Génomes, Comportement, Ecologie, CNRS, IRD, Univ. Paris-Sud, Université Paris-Saclay, F-91198 Gif-sur-Yvette, France. [3] AgroParisTech, Université Paris-Saclay, F-75005 Paris, France. Correspondence and requests for materials should be addressed to R.B. (email: Richard.Benton@unil.ch).

For humans, food texture is a much-marvelled aesthetic property, encompassing diverse qualities such as hardness, brittleness, chewiness and gumminess[1]. For all animals, however, texture provides important, and often vital, information on how easy food will be to masticate, swallow and digest, as well as warning about products succumbing to decomposition by pathogenic microbes. While our oral assessment of food texture by the tongue and palate is very sensitive—uncooked vegetable fibres or pieces of gristle are readily detected in a mouthful—remarkably little is known about the molecular and cellular mechanisms by which food texture is sensed in any animal.

*Drosophila melanogaster* provides an appealing model system to investigate the mechanisms of food texture assessment, as the feeding preferences of flies can be assessed through simple choice assays, and electrophysiological analyses[2] and powerful neurogenetic manipulations[3] can be used to ascribe sensory functions to specific receptors and neural pathways. Gustatory assessment of food by flies is well-described for chemical components of food[4], including appetitive and aversive compounds (for example, sugars and bitter toxins, respectively). Here, members of the Gustatory Receptor (Gr) or Pickpocket (Ppk) families of sensory ion channels mediate tastant-evoked neuronal activity and behaviour[4]. By contrast, how *Drosophila* discriminates food's textural properties has been little studied.

Here we show that flies show robust and sensitive discrimination of food substrates of distinct textures, and identify a mechanosensory ion channel and neural population that are important for this sensory-guided feeding behaviour.

## Results

**Drosophila can discriminate substrate texture**. To determine whether substrate texture influences *Drosophila* feeding behaviour, we first adapted a two-choice assay in which flies (starved before the experiment) are allowed to feed *ad libitum* on a microtitre plate in which alternate wells contain sucrose dissolved in different concentrations of agarose (Fig. 1a). Linear agarose chains aggregate during gelation to form a three-dimensional mesh of channels, whose diameter (10s to 100s nm, depending on polymer concentration) influences textural properties, including hardness, viscosity and springiness[5]. We reasoned that this controlled assay would provide an approximation of at least some elements of the choices flies make in nature on foods of different textures. We tested preference for substrates composed of 0.5% or 2% agarose; these contained distinct edible dyes to permit assessment of the feeding of flies by scoring the coloration of their abdomens. This experiment revealed a strong preference for feeding from the softer agarose (Fig. 1a). Using Semmes–Weinstein Monofilaments (see Methods), we estimated that the hardness of 0.5% agarose is within the range of that of the flesh or damaged skin of a variety of fruits, where *Drosophila* feed[6], while 2% agarose is more similar to the undamaged skin of ripe fruit (Fig. 1b). These observations suggest that the agarose concentrations resemble conditions that flies need to discriminate in the wild.

We extended this initial result by developing a preference assay in which we recorded the position of freely roaming flies between substrates containing the same concentration of sucrose but different concentrations of agarose, within a four-quadrant circular arena (Fig. 1c). This assay is advantageous because it provides a more natural test of food source assessment by allowing flies to walk directly on the substrate, offers higher resolution information on the temporal development of any preferences, and avoids the potential confounding influences of innate chemosensory preferences for the food dyes. Consistent with our previous experiment, flies preferentially accumulated on 0.5% agarose when compared with 2% agarose. We expressed this choice as a preference index (PI) for 0.5% agarose, which increased over time until a plateau of ∼0.7 was reached after 60 min (reflecting ∼85% of flies on this substrate) (Fig. 1d).

Using this assay, we compared the preference between 0.5% agarose and a range of other substrate concentrations. Softer agarose (0.25%) was slightly preferred to 0.5%; however, at a lower agarose concentration (0.02%), flies preferred 0.5% agarose (Fig. 1e), potentially because of difficulty in walking on a substrate of near-liquid consistency.

**Flies discriminate texture when feeding**. In nature, flies can be found on various non-food surfaces (for example, wood, rock) with diverse textures, suggesting that texture preference may only be exhibited when flies are searching for or consuming food. In support of this idea, we found that flies that had not been starved displayed no preference for 0.5% over 2% agarose in the presence of sucrose (Fig. 2a; Supplementary Fig. 1a). Moreover, we found that starved flies' positional preference for 0.5% over 2% agarose was greatly diminished in the absence of sucrose (Fig. 2b, left and Supplementary Fig. 1b), which correlated with a substantial reduction in the percentage of flies feeding on these plates, as assessed by their ingestion of a food dye mixed in the agarose (Fig. 2b, right). Replacement of sucrose with sorbitol, which is a nutritious but 'tasteless' sugar for *Drosophila*[7,8] lead to expression of little, if any, texture preference (Fig. 2c), suggesting that stimulation of peripheral appetitive sensory pathways is necessary to reveal texture preference.

We tested this hypothesis by selectively inhibiting different chemosensory pathways and determining how this influenced flies' preference for softer agarose in the presence of sucrose (Fig. 2d; Supplementary Fig. 1c). Blockage of sugar-sensing neurons (with a Gr64f-Gal4 (ref. 9) driver inducing a UAS-Tetanus toxin (TNT) transgene) abolished preference, compared with a control line expressing an impaired version of TNT (TNT[IMP]) (Fig. 2d; Supplementary Fig. 1c). Inhibition of water-sensing neurons (using Ppk28-Gal4 (ref. 10)) also led to diminished, but not abolished, preference for 0.5% agarose (Fig. 2d; Supplementary Fig. 1c). By contrast, in control animals in which chemosensory neurons for aversive stimuli were inhibited (using Gr66a-Gal4 (ref. 11)), we saw no effect on texture preference (Fig. 2d; Supplementary Fig. 1c). These results indicate that peripheral sensory detection of sugar and, to some degree, water, is necessary for texture discrimination.

To eliminate the possibility that flies' preference for softer agarose is simply because of the greater accessibility of sucrose in this less dense substrate, we performed two further experiments. First, we prepared arenas with 0.5% and 2% agarose quadrants but without sugar; we then applied uniformly on the surface of the agarose a concentrated sucrose solution, which was allowed to be absorbed before testing flies' preference. If sucrose diffuses less quickly in the denser agarose and remains accessible on the surface, we reasoned the perceived sucrose concentration would be higher in the 2% agarose quadrants. However, flies still displayed a strong preference for 0.5% agarose (Fig. 2e). In the second experiment, we prepared 0.5% versus 2% agarose arenas without sucrose and optogenetically activated sweet-sensing neurons throughout all quadrants with the red-shifted channel-rhodopsin CsChrimson[12]. These flies display substantially stronger preference for 0.5% agarose compared with control strains containing only Gr64f-Gal4 or UAS-CsChrimson transgenes (Fig. 2f). Thus, while appetitive stimuli are important for flies to display texture discrimination, this behaviour is not because of differences in availability or detection of appetitive chemicals.

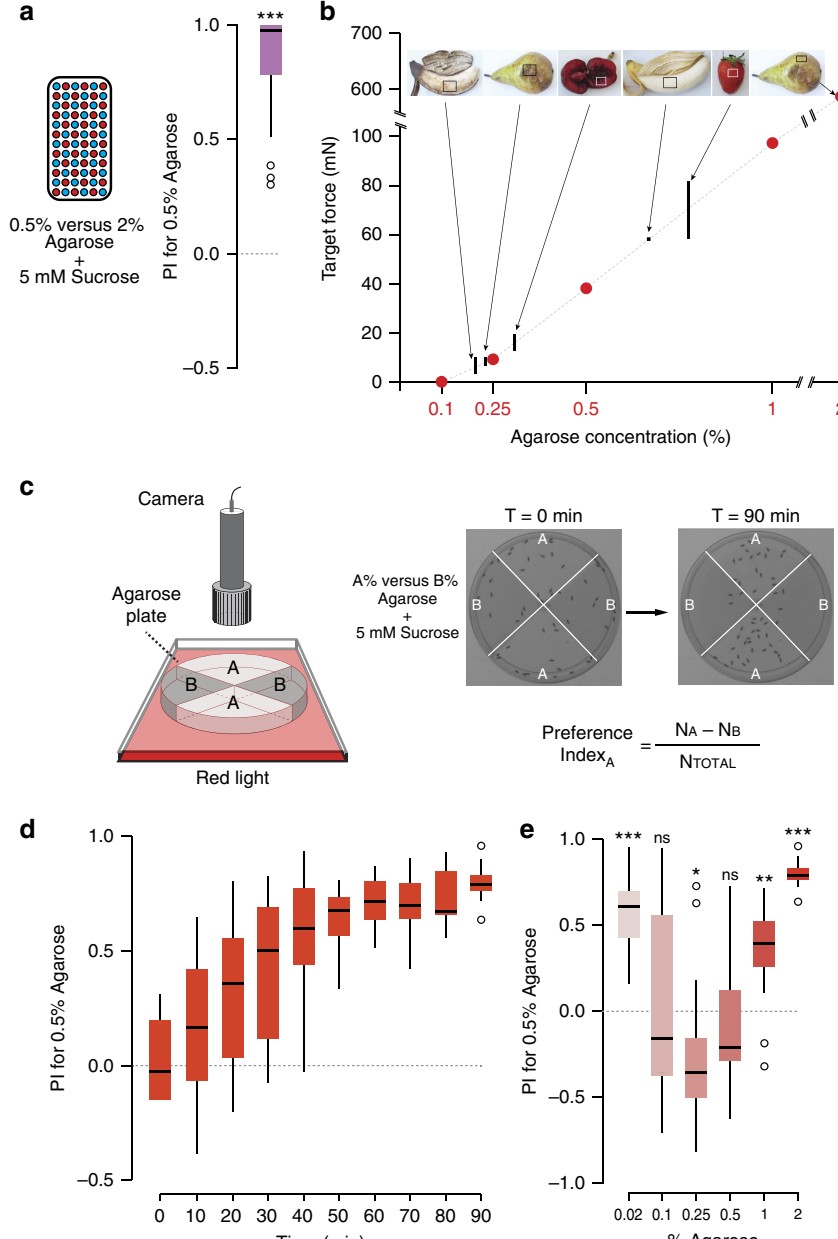

**Figure 1 | Flies display feeding and positional preference for substrate texture. (a)** Left: schematic of the two-choice colorant feeding assay. Flies can choose to feed from 5 mM sucrose in substrates composed of 0.5 or 2% agarose coloured with different edible dyes (which are switched in different trials) in alternate wells of a microtitre plate. Feeding preference is determined by abdominal coloration of flies after 90 min. Right: PI of wild-type ($w^{1118}$) flies for feeding from 0.5% agarose substrates, calculated as described in the Methods ($n = 20$ experiments). ***$P < 0.001$ (Wilcoxon signed rank test ($H_0 = 0$)). **(b)** Graph of substrate stiffness (target force in milliNewtons measured using Semmes-Weinstein Monofilaments; see Methods) of substrates of different agarose concentrations (red dots). Overlaid are the ranges of similar measurements made from the flesh or skin of the fruit samples shown in the photos (Fujifilm XT1 camera; 18–55 mm objective); compared with agarose, natural food substrates have heterogeneous properties. **(c)** Left: schematic of two-choice positional preference arena assay. Flies can choose to feed from 5 mM sucrose in substrates composed of different agarose concentrations (without dyes) in alternate chambers of a 90 mm diameter four-quadrant plate. Right: representative images of flies in the assay arena at the start and end of an experiment. Fly position was quantified automatically and used to calculate a Preference Index (PI) as indicated below the images. **(d)** Time course of PI of wild-type flies for 0.5% agarose over 2% agarose ($n = 12$ arenas). **(e)** Preference of wild-type flies for 0.5% agarose in the arena assay (at time = 90 min, in this and subsequent assays unless otherwise stated) with different alternative substrate agarose concentrations; all substrates contain 5 mM sucrose ($n = 15$ for 0.02% agarose, $n = 18$ for 0.1% agarose, $n = 14$ for 0.25% agarose, $n = 15$ for 0.5% agarose, $n = 15$ for 1% agarose, $n = 12$ for 2% agarose). ns: not significant, ***$P < 0.001$, **$P < 0.01$, *$P < 0.05$ (Wilcoxon signed rank test ($H_0 = 0$)).

**NOMPC is required for texture discrimination.** In humans, vision contributes to initial assessment of texture (for example, surface properties)[1]. Although agarose quadrants of distinct densities have slightly different appearances (for example, Fig. 1c), we could eliminate the contribution of visual cues for *Drosophila*, because all assays were performed in conditions

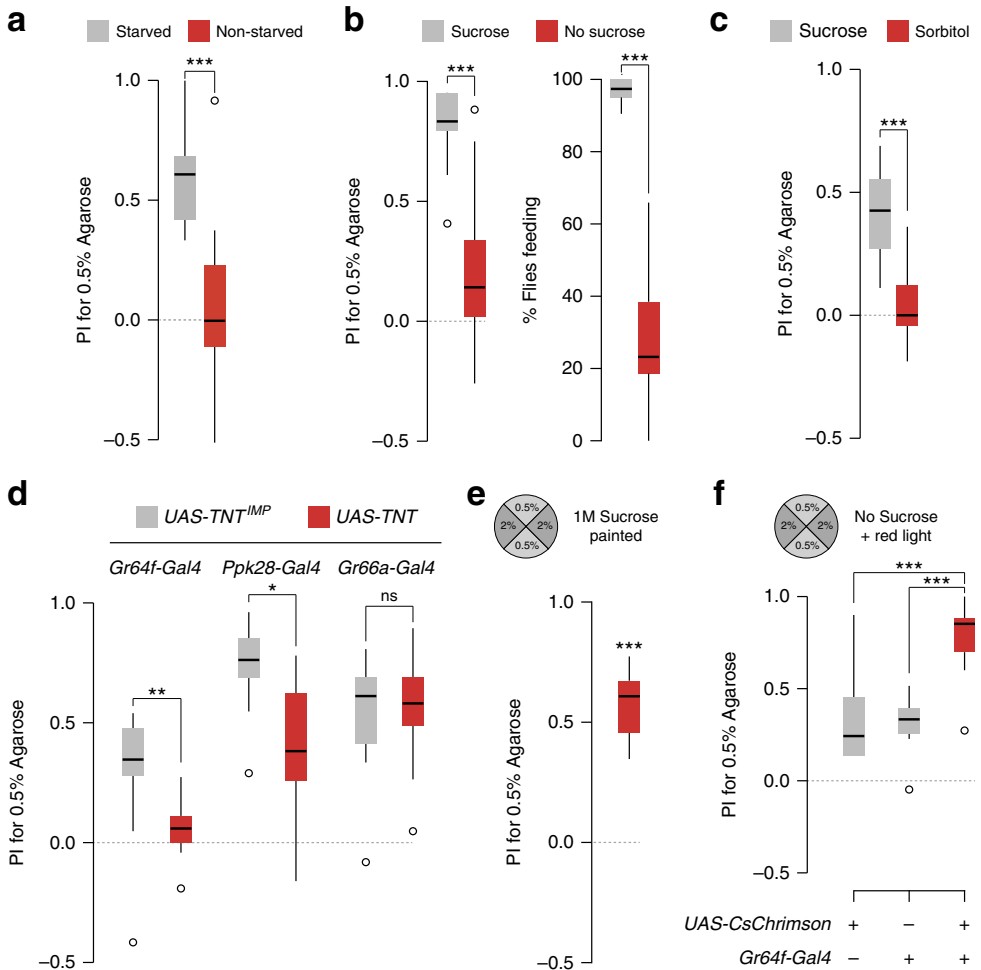

**Figure 2 | Appetitive gustatory signals are necessary and sufficient to reveal texture preference.** (**a**) Preference of starved (grey) or non-starved (red) wild-type flies for 0.5% agarose in a 0.5% versus 2% assay arena in the presence of sucrose ($n = 12$ for both starved and non-starved conditions). ns: not significant, ***$P < 0.001$ (Wilcoxon rank sum test). (**b**) Left: Preference of wild-type flies for 0.5% agarose in a 0.5% versus 2% arena assay in the presence (grey) or absence (red) of 5 mM sucrose ($n = 13$ with sucrose, $n = 14$ without sucrose). Right: Percentage of flies feeding in the arena assay in the presence (grey) or absence (red) of 5 mM sucrose in parallel assays containing blue food dye in all quadrants, as determined by abdominal coloration of animals after 90 min ($n = 13$ with sucrose, $n = 14$ without sucrose). ***$P < 0.001$ (Wilcoxon rank sum test). (**c**) Preference of wild-type flies for 0.5% agarose in a 0.5% versus 2% arena assay in the presence of 5 mM sucrose (grey) or 5 mM sorbitol (red) ($n = 12$ for sucrose, $n = 13$ for sorbitol). ***$P < 0.001$ (Wilcoxon rank sum test). (**d**) Preference for 0.5% agarose in a 0.5% versus 2% arena assay (with 5 mM sucrose) of flies in which Tetanus toxin (TNT) is expressed under the control of *Gr64f*, *Ppk28* or *Gr66a* promoters. Comparisons are made with control animals expressing an impaired version of this toxin (TNT$^{IMP}$). The PI at time $= 120$ min is shown, because animals bearing the *Gr64f-Gal4* transgene display delayed decisions; the temporal evolution of PI is shown in Supplementary Fig. 1c. Genotypes are: *w;Gr64-Gal4f/UAS-TNT* ($n = 14$), *w;Gr64f-Gal4/UAS-TNT$^{IMP}$* ($n = 12$), *w;Ppk28-Gal4/UAS-TNT* ($n = 8$), *w;Ppk28-Gal4/UAS-TNT$^{IMP}$* ($n = 8$), *w;Gr66a-Gal4/UAS-TNT* ($n = 13$), *w;Gr66a-Gal4/UAS-TNT$^{IMP}$* ($n = 12$), ns: not significant, **$P < 0.01$, *$P < 0.05$ (Wilcoxon rank sum test). (**e**) Preference of wild-type flies for 0.5% agarose in a 0.5% versus 2% arena assay in which 100 μl of 1 M sucrose solution is applied uniformly on the surface of each quadrant and allowed to dry before introducing the flies ($n = 12$). ***$P < 0.001$ (Wilcoxon signed rank test ($H_0 = 0$)). (**f**) Preference of flies for 0.5% agarose in a 0.5% versus 2% arena assay (without sucrose), in which *Gr64f* sweet-sensing neurons are uniformly activated across all quadrants through optogenetic stimulation. Genotypes: *w;Gr64f-Gal4/Gr64f-Gal4* ($n = 14$); *w;UAS-CsChrimson/UAS-CsChrimson* ($n = 16$); *w;Gr64f-Gal4/UAS-CsChrimson* ($n = 19$). ***$P < 0.001$ (Wilcoxon rank sum test).

where flies have no visual input. Reasoning that touch was the most likely responsible sensory modality, we screened a number of known mechanosensory channels and found that animals mutant for *no mechanoreceptor potential C* (*nompC*)[13] had severely disrupted texture preference (Fig. 3a; Supplementary Fig. 2a). Importantly, while *nompC* null mutants have locomotor defects[14] (and were excluded from our behavioural analyses), the hypomorphic *nompC* allelic combinations tested displayed no obvious defects in locomotion and exploration of the assay arena, or in texture-independent preference for sucrose (Supplementary Fig. 2b and c). These results indicate that the defect in texture

discrimination of *nompC* mutants is likely to reflect a direct role in this sensory modality.

**NOMPC localizes to specific labellar sensilla neuron cilia.** NOMPC is a Transient Receptor Potential family ion channel that functions in sensory organs involved in proprioception[14], gentle touch detection[15] and hearing[16,17]. Given our observation that texture preference is only revealed when flies are feeding, we asked whether NOMPC might also be expressed in the labellum, the distal end of the proboscis that directly contacts substrates

when flies are assessing potential food sources and eating. Using a NOMPC antibody, we detected this channel in single neurons associated with individual labellar sensilla—the sensory hairs that house gustatory neurons—with the protein concentrating in the ciliated sensory dendrite at the base of the cuticle (Fig. 3b). This signal is largely or completely abolished in hypomorphic or complete loss-of-function alleles of *nompC*, confirming the specificity of the antibody (Fig. 3b). These neurons are very likely to correspond to those suggested to be mechanosensory based on morphological features[18]. Consistently, the NOMPC-positive dendrites are distinct from those labelled by reporters for appetitive- or aversive-stimulus sensing gustatory neurons (Fig. 3c).

To examine the anatomical projections of these neurons, we tested *nompC*-promoter driver lines. A previously-characterised *nompC-Gal4* line[14], containing 1.5 kb of 5′ regulatory sequence, does not label labellar neurons (Fig. 3d, left), although it is strongly expressed in auditory mechanosensory neurons that innervate the antennal mechanosensory motor centre (AMMC)[16,17] (Fig. 3e, left). However, a *nompC-LexA* driver[19], containing ∼9.5 kb regulatory sequence spanning the translation start site, is expressed robustly in NOMPC-positive labellar neurons (Fig. 3d, right). Essentially all labellar sensilla examined were innervated by a single *nompC-LexA > CD8:GFP* expressing neuron (136/142 sensilla in eight labella; in 6 sensilla we observed dendritic CD8:GFP signals but could not unambiguously link these to a soma). We found no overlap in the expression of this driver with those for sweet, bitter or water taste neuron classes (Supplementary Fig. 3a, top), indicating that *nompC-LexA* comprehensively and selectively labels the labellar mechanosensory neuron population. In the SEZ, these neurons

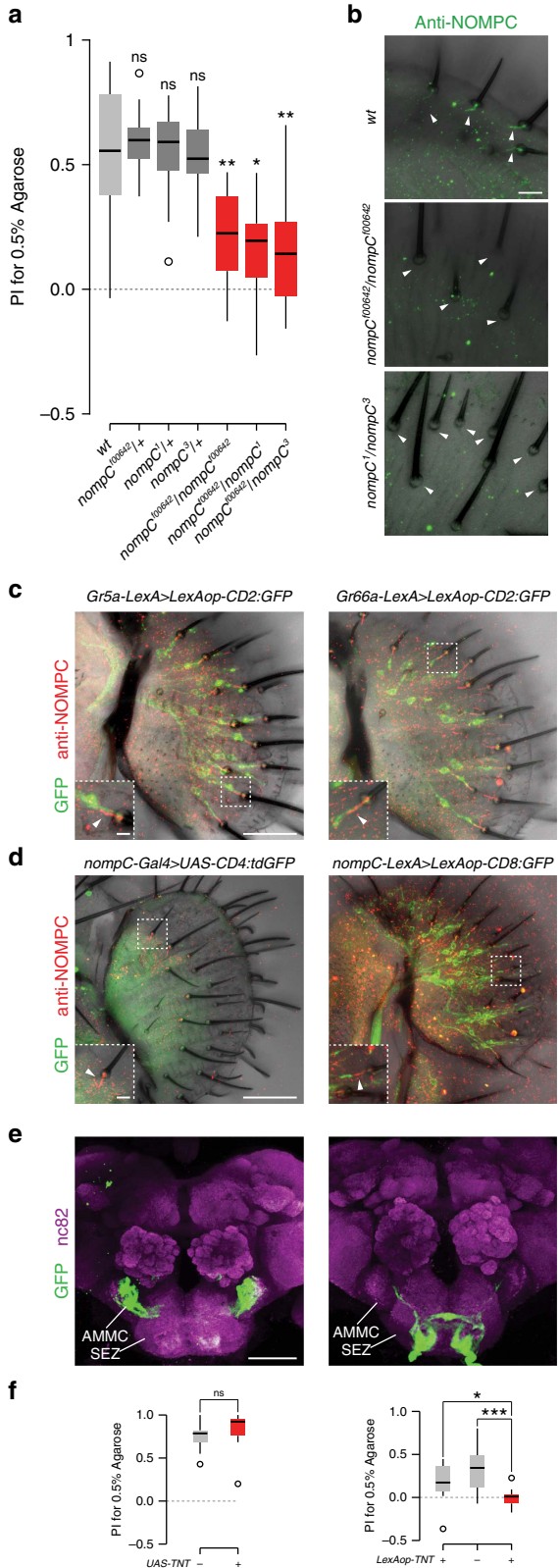

**Figure 3 | The mechanosensory channel NOMPC is necessary for texture discrimination and expressed in gustatory sensilla neurons.**
(**a**) Preference of the indicated control and *nompC* mutant flies for 0.5% agarose in a 0.5% versus 2% arena assay (with 5 mM sucrose). Wild-type ($n = 17$), $nompC^{f00642}/+$ ($n = 18$), $nompC^{1}/+$ ($n = 15$), $nompC^{3}/+$ ($n = 15$), $nompC^{f00642}/nompC^{f00642}$ ($n = 17$), $nompC^{f00642}/nompC^{1}$ ($n = 10$), $nompC^{f00642}/nompC^{3}$ ($n = 12$). ns: not significant, **$P < 0.01$, *$P < 0.05$. Comparisons were made against the wild-type control (Wilcoxon rank sum test with Bonferroni correction for multiple comparisons). (**b**) Immunofluorescence with anti-NOMPC (green, on a bright-field background) on whole-mount labella. In wild-type animals, NOMPC concentrates in the distal tip of sensory neurons that terminate at the base of each taste sensillum (arrowheads); *nompC* mutants lack this expression (arrowheads). Scale bar, 10 μm. (**c**) Immunofluorescence with anti-NOMPC (red) on whole-mount labella of animals in which different gustatory sensory neurons (GSNs) are transgenically labelled (green). Genotypes: *w;Gr5a-LexA,LexAop-rCD2:GFP;TM2/TM6B*, *w;Gr66a-LexA,LexAop-rCD2:GFP;TM2/TM6B*. Scale bar, 50 μm. Inset: NOMPC does not colocalise with GSNs (arrowheads). Scale bar, 25 μm. Green and red colocalisation is sometimes apparent in the full-projection because of vertical superposition of the NOMPC-labelled and GSN sensory processes. (**d**) Immunofluorescence with anti-NOMPC (red) and anti-GFP (green) on whole-mount labella of animals of the indicated genotypes (*w;UAS-CD4:tdGFP;nompC-Gal4* and *w;LexAop-CD8:GFP-2A-CD8:GFP; nompC-LexA*). Scale bar, 50 μm. Inset: NOMPC expression is adjacent to the GFP (arrowheads). The *nompC-Gal4* line is not expressed in the labellum (arrowheads). Scale bar, 25 μm. (**e**) Immunofluorescence with anti-GFP (green) and nc82 (magenta) on whole-mount brains of animals (genotypes as in **d**). The weak green signal in the SEZ in *nompC-Gal4* animals does not originate from labellar neurons. SEZ: Subesophageal Zone; AMMC: Antennal Mechanosensory and Motor Center. Scale bar, 50 μm. (**f**) Preference for 0.5% agarose in a 0.5% versus 2% arena assay (with 5 mM sucrose) of animals in which different *nompC* neuron populations are silenced, together with control lines. Left: genotypes: *w;nompC-Gal4/UAS-TNT^{IMP}* ($n = 10$) and *w;nompC-Gal4/UAS-TNT* ($n = 10$). ns: not significant (Wilcoxon rank sum test). Right: genotypes: *w;nompC-LexA/LexAop-TNT* ($n = 12$), *w;nompC-LexA/+* ($n = 16$), *w;LexAop-TNT/+* ($n = 12$). ***$P < 0.001$, *$P < 0.05$ (Wilcoxon rank sum test with Bonferroni correction for multiple comparisons).

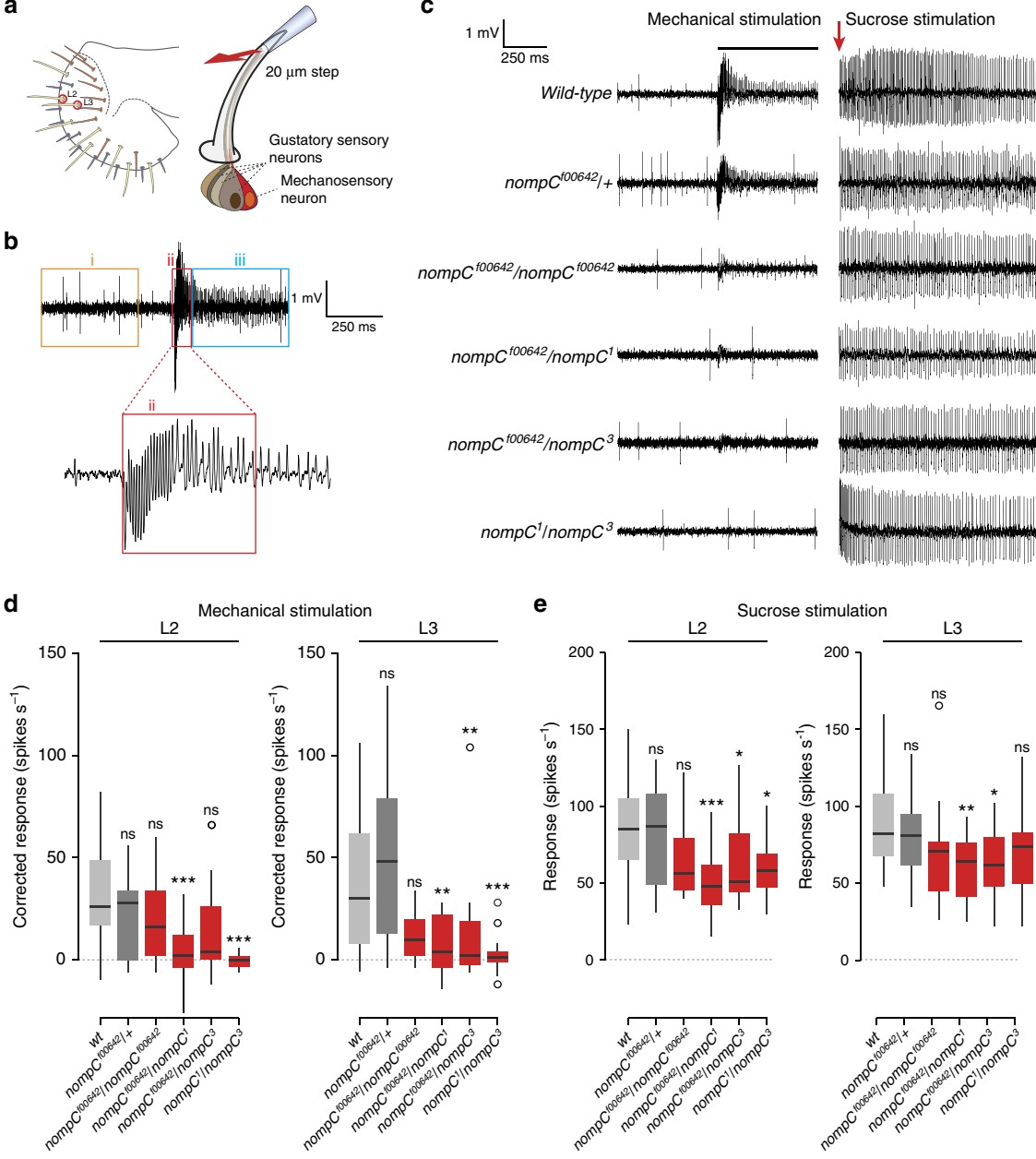

**Figure 4 | Gustatory sensilla display NOMPC-dependent mechanosensory responses.** (**a**) Left: schematic of the *Drosophila* labellum showing the distribution and diverse orientations of different morphological classes of sensilla, including the 'long' L2 and L3 classes[20] examined in this study. Right: schematic of the cellular organization of L2 and L3 sensilla, which house four GSNs and a single presumed mechanosensory neuron (red) whose dendrites terminate at the base of the cuticular hair. For mechanosensory stimulation, the hair is displaced 20 μm by Piezo-controlled movement of a glass recording pipette placed over the end of the sensillum (see Methods). (**b**) Representative electrophysiological trace of an L3 sensillum before, during and after mechanosensory stimulation. (i) Before stimulation, basal neuronal spikes of varying amplitudes are visible; large spikes are likely to correspond to the sweet-sensing neuron, and smaller spikes to the other neurons in this sensillum. (ii) during movement of the hair, there is a negative deflection of the base-line and a fast train of larger 'spike-like' electrical oscillations is detected, which lasts ~100 ms. (iii) This initial electrical response eventually resolves into a series of small amplitude spikes. For quantifications, see Methods. (**c**) Representative traces of responses of L3 sensilla to mechanical stimulation and 100 mM sucrose stimulation in the indicated control and *nompC* mutant genotypes. The horizontal bar indicates the period of hair bending; the arrow indicates the time of contact of the sucrose-containing pipette with the sensillum. (**d** and **e**) Neuronal responses of L2 and L3 sensilla to mechanical stimulation (**d**) and to 100 mM sucrose (**e**) in the indicated genotypes. Wild-type ($n_{L2} = 31$, $n_{L3} = 33$), *nompC*[f00642]/+ ($n_{L2} = 17$, $n_{L3} = 19$), *nompC*[f00642]/*nompC*[f00642] ($n_{L2} = 17$, $n_{L3} = 17$), *nompC*[f00642]/*nompC*[1] ($n_{L2} = 17$, $n_{L3} = 17$), *nompC*[f00642]/*nompC*[3] ($n_{L2} = 17$, $n_{L3} = 16$), *nompC*[1]/*nompC*[3] ($n_{L2} = 14$, $n_{L3} = 16$). All comparisons were made against the wild-type control. ns: not significant, \*\*\*$P < 0.001$, \*\*$P < 0.01$, \*$P < 0.05$ (Wilcoxon rank sum test with Bonferroni correction for multiple testing). Although the weak hypomorphic *nompC*[f00642] allele shows decreased responses, these are not statistically different from controls; the behavioural phenotype of this mutant (Fig. 3a) may reflect the consequence of the combination of sensory defects across multiple sensilla.

innervate a ventral region (Fig. 3e, right); these projections are discrete from those of any classes of gustatory neuron, except for a small zone of overlap with sweet neurons (Supplementary Fig. 3a, bottom). Unexpectedly, *nompC-LexA* does not label sensory neurons innervating the AMMC (Fig. 3e, right), indicating the existence of modular activity of the *nompC* promoter in different sensory structures. Differential *nompC* driver expression is also observed in neurons innervating the ventral nerve cord (Supplementary Fig. 3b).

We took advantage of these drivers to examine the role of different populations of *nompC* neurons in texture detection. Expression of TNT in neurons labelled by *nompC-Gal4*—which comprise auditory neurons in the head, and chordotonal and mechanosensory bristle neurons that project to the ventral nerve cord[14]—did not affect texture preference (Fig. 3f, left). By contrast, TNT expression under the control of *nompC-LexA* abolished texture discrimination (Fig. 3f, right), implicating labellar neurons in this behaviour.

**NOMPC-dependent mechanosensory activity of labellar neurons**. Although morphological studies have suggested the existence of a mechanosensory neuron in gustatory sensilla[18], physiological investigation of this possibility has, to our knowledge, never been performed in *Drosophila*. We adapted the tip-recording methodology for measuring tastant-evoked responses in chemosensory neurons[2] for mechanosensory stimulation and recordings (Fig. 4a and Methods). Small (20 μm) physical displacements of labellar sensilla—focussing on the easily-accessible L2 and L3 sensillar classes[20] (Fig. 4a)—with the recording pipette resulted in robust trains of action potentials (Fig. 4b), demonstrating that labellar sensilla are able to convey mechanosensory, and not only gustatory, signals. Importantly, in *nompC* strong hypomorphic and null mutants, these mechanosensory-evoked neuronal responses were reduced or completely abolished (Fig. 4c,d). By contrast, responses of the gustatory sensory neuron to sucrose in these sensilla were readily detected in all genotypes (Fig. 4c,e). In certain strains, responses showed statistically-significant reductions, but this phenotype did not correlate with the strength of the *nompC* mutant allele suggesting it is because of genetic background influences. These results indicate that NOMPC has a specific role in mechanotransduction in labellar gustatory sensilla neurons.

## Discussion

This work provides evidence for a NOMPC-dependent mechanosensory pathway that contributes to discrimination of food textural properties by *Drosophila*. While NOMPC has numerous functions in different sensory appendages, our experiments implicate labellar mechanosensory neurons as key for food texture assessment. NOMPC-dependent leg mechanosensory neurons[21]—that, similar to labellar sensilla, are likely to be grouped together with taste neurons[22,23]—may also be an important source of information on textural properties. However, the observation that texture preference (under our assay conditions, at least) is exhibited only when flies are starved and actually feed from (rather than simply walk on) appetitive substrates supports an important role for the labellum in detecting texture stimulus information that informs feeding decisions.

Our electrophysiological demonstration that chemosensory sensilla on the labellum each house a NOMPC-dependent mechanosensory neuron illustrates an elegant way in which food texture can be assessed: probing of the food surface by the proboscis is likely to produce physical bending of labellar hairs and activation of the mechanosensitive neurons. For a constant

placement of the labellum relative to the substrate, increased substrate hardness will presumably lead to increased bending of sensilla, potentially offering quantitative information to the fly about the stiffness of the food source. It remains unclear how flies integrate the textural and chemosensory qualities of food. The compartmentalization of gustatory and mechanosensory neurons within a common sensillum opens the possibility for non-synaptic inhibitory interactions, as occurs in paired olfactory sensory neurons in the antenna[24]. In such a scenario, a hard, sweet substrate would be less appetitive than a soft, equally-sweet substrate, because the former would evoke higher mechanosensory neuron firing that would lead to greater inhibition of the sweet taste neuron. It is also likely that integration occurs within the SEZ and/or higher brains centres. Recent work has suggested that labellar mechanosensory neurons are GABAergic and, when optogenetically-activated, can inhibit sweet neuron activity, potentially through direct contact[25]. We see only very limited overlap between mechanosensory and sweet neuron projections in the SEZ (which may or may not reflect synaptic contacts) making it likely that additional mechanisms facilitate multisensory integration, such as still-poorly understood SEZ interneurons that represent chemical and mechanical signals in the brain[26].

Given the diversity in physical properties of substrates that flies feed on, as well as the importance of textural assessment for other behaviours such as walking and oviposition (where flies must avoid overly liquid substrates in which they would risk becoming stuck and/or drowning), additional neural and molecular mechanisms are undoubtedly involved. The recent characterization of a pair of force-activated multidendritic neurons expressing transmembrane channel-like protein in the labellum[27] provides one important example. The striking anatomical properties of these neurons suggest that they integrate mechanical forces across the surface of the labellum, complementary to the NOMPC mechanosensory neurons that transduce local information from each sensillum. A future challenge in understanding texture detection will be to identify all of the relevant sensory pathways, develop behavioural paradigms to finely distinguish their individual roles, and delineate how these pathways integrate in the brain.

## Methods

**Drosophila strains.** *Drosophila* stocks were maintained on a standard corn flour, yeast and agar medium under a 12 h light:12 h dark cycle at 25 °C. The wild-type strain was *w1118*. Other mutant and transgenic strains were: *nompCf00642* (hypomorphic allele)[28], *nompC1*, *nompC3* (null alleles)[13], *nompC-Gal4* (ref. 14), *nompC-LexA* (ref. 19), *Gr66a-Gal4* (ref. 11), *Gr64f-Gal4* (ref. 9), *Ppk28-Gal4* (ref. 10), *Gr5a-LexA* (ref. 29), *Gr66a-LexA* (ref. 30), *UAS-CsChrimson*[12], *UAS-TNT*, *UAS-TNTIMP* (*IMPTNT-V*)[31], *UAS-CD4:tdGFP* (ref. 32), *LexAop-CD8:GFP* (ref. 33), *LexAop-CD2:GFP* (ref. 33), *UAS-CD4:tdTomato*[32], *13xLexAop2IVS-TNT-HA* (ref. 34).

**Behaviour.** All behaviour assays were performed with 3–5 day old males in a dark room with controlled temperature (25 °C) and relative humidity (60%). Flies of the desired genotype were selected and kept in vials with fresh food for a minimum of one day. Before experiments, flies were starved for 24 h in glass tubes containing a Kimwipe soaked in 2 ml tap water. Non-starved flies were placed in tubes with new food for 24 h before the experiment. Flies were ice anesthetised before transfer to assay arenas. Behavioural arenas were prepared in the morning and experiments were performed in the afternoon. Control and test genotypes and different assay conditions for all experiments were run in parallel.

*Two-choice colorant feeding assay.* Assays were performed adapting a previous protocol[35]. Arenas consisted of 72-well microtitre plate (Sigma M5812) in which wells were filled alternately with 20 μl 5 mM sucrose (Sigma S5390) in 0.5% or 2% agarose (Promega V3125), dyed with blue or red food dyes (Food Blue No. 1 and Food Red No. 106 dyes; Tokyo Chemical Industry Co. (Tokyo, Japan)) (Fig. 1a). Dyes were exchanged in different experiments to avoid any bias because of flies' gustatory discrimination of these chemicals. Approximately 50–60 flies were introduced into an arena and allowed to feed for 90 min after which plates

were placed at $-20\,°C$, and flies recovered to score the number (#) with Blue (B), Red (R) or Purple (P) abdomens. PI for 0.5% agarose was calculated as: $(\#_B + 0.5\#_P)/(\#_R + \#_B + \#_P)$ or $(\#_R + 0.5\#_P)/(\#_R + \#_B + \#_P)$, depending on the % agarose/dye combination.

*Two-choice positional preference arena assay.* Arenas consisted of 90 mm diameter four-section plastic plates (Phoenix Biomedical 332), in which quadrants were alternately filled with two different concentrations of agarose (11.5 ml per quadrant) (Fig. 1c). Unless indicated otherwise, all quadrants contained 5 mM sucrose. 40–50 flies were introduced into each arena, and up to 16 arenas were placed on a methacrylate panel (1.5 cm thickness) elevated 5.5 cm from the light source (a $60 \times 60$ cm LED Panel (Ultraslim LED Panel, 360 Nichia LEDs, Lumitronix) covered with red film (106 Primary Red, Showtec). Pictures were taken (using a USB 3.0 1″ CMOS Monochrome Camera $2048 \times 2048$ Pixel and a CCTV Lens for 2/3″f:16 mm (iDS)) at different time points for a maximum of 90 min, or 120 min for the experiments in Fig. 2d; Supplementary Fig. 1c. The distribution of flies in the arena was quantified using a custom macro in ImageJ (available upon request). Preference index for 0.5% agarose was calculated as: $(N_{\text{flies\_0.5\%\_agarose}} - N_{\text{flies\_x\%\_agarose}})/N_{\text{total\_flies}}$, where $x$ represents the concentration in the complementary quadrants. For the experiment in Fig. 2b, a blue food dye (brilliant blue FCF, $0.125$ mg ml$^{-1}$; Sigma-Aldrich 027-12842) was added to all agarose quadrants. For the experiment in Fig. 2c, 5 mM sorbitol (Sigma S6021) was used instead of sucrose. For the experiment in Fig. 2e, 100 μl 1 M sucrose was spread over each agarose quadrant using a glass spreader until the liquid was fully absorbed; flies were introduced immediately after to minimize further diffusion of surface-applied sucrose into the agarose.

*Quantification of exploratory/locomotor activity.* Assays were performed in the same arenas as for the two-choice positional preference assay with 0.5% versus 2% agarose quadrants plus 5 mM sucrose. Approximately 10–20 starved males were placed in each plate and their movements video-recorded at 2 fps for 5 min. To quantify the area covered by the wild-type and *nompC* mutant flies we focussed on only the middle 120 frames (during the third minute) to minimize overlap of individual animal's trajectories, which confound accurate quantifications. All the frames to be quantified were projected (using minimal projection in ImageJ) into a single image that contains the positions of all flies throughout this period. The background was subtracted and the image thresholded so each fly was represented as a set of black pixels. The area covered by each fly was calculated by dividing the number of black pixels by the total area of the plate and by the number of flies on the plate. Dead flies found in the plate (and the area they covered) were not considered in the analysis.

*Optogenetic stimulation.* Flies were raised in the dark in standard fly food supplemented with 1.5 mM all-*trans*-retinal (Sigma-Aldrich R2500). Selected males were placed in tubes containing fresh food with retinal for a minimum of one day. Agarose plates (without sucrose) were inverted to ensure homogeneous light stimulation (that is, through the lid), using the same light source as described above (4.2–5.5 μW mm$^{-2}$, measured at 633 nm using a PM200 Thorlabs power metre with a $18 \times 18$ mm$^2$ S170C sensor).

**Substrate stiffness measurements.** To compare the stiffness of substrates of different agarose concentration and various fruits, we used the Semmes–Weinstein Monofilament set (Bioseb). When an individual filament is applied against a surface, the force to which it is exposed increases until it penetrates the surface or, if the surface resists penetration, until the filament bends (after which the force no longer increases). The filament set consists of a range of diameters and lengths, which are calibrated such that the force necessary to induce bending is known; the set provides an approximate logarithmic scale of actual force applied. Successive filaments were applied to the surface of interest at least five times each, starting with the lowest 'target force' (thinnest) filament, until we identified the first that was able to penetrate without bending. In contrast to agarose, for some fruits we observed heterogeneity in stiffness within the tested area (Fig. 1b); in these cases we plotted the range of target forces measured.

**Immunohistochemistry and imaging.** Immunofluorescence on whole-mount labella was performed adapting a protocol for whole-mount antennae[36]. In brief, proboscides were dissected and fixed for 20 min in 4% PFA in PBS + 0.2% Triton X-100 (PBT) at $4\,°C$. All washes were performed at least three times in PBT at room temperature. Primary and secondary antibody incubations were for 48 h each in PBT + 5% inactivated goat serum at $4\,°C$. Immunofluorescence on whole-mount brains was performed following a standard protocol[37], except that flies were fixed for 3 h at $4\,°C$. Primary antibodies: rabbit anti-NOMPC (ref. 14), mouse monoclonal nc82 (diluted 1:10; Developmental Studies Hybridoma Bank), chicken anti-GFP (1:1,000; Abcam), rabbit anti-RFP (1:1,000; Abcam). Secondary antibodies: Alexa488- and Cy3-conjugated goat anti-rabbit or anti-mouse IgG, respectively (1:100; Molecular Probes and Jackson ImmunoResearch), goat anti-chicken Alexa488 (1:100; Abcam) or goat anti-mouse Cy5 (1:100; Jackson ImmunoResearch). Microscopy was performed using an LSM 710 laser scanning confocal microscope (Zeiss) and images were processed with ImageJ.

**Electrophysiology.** For all electrophysiological experiments, 1–2 day old flies were used. *nompC$^1$/nompC$^3$* mutant flies show severe incoordination and die soon after

hatching, typically because they become stuck in culture medium (note: this allelic combination was not used for behavioural experiments). However, we were able to obtain young adults by transferring pupae to culture tubes containing a Kimwipe moistened with 3 ml 5 mM sucrose and recovering flies soon after eclosion. For consistency, all genotypes were treated in this manner for analysis. Genotypes were interleaved to minimize effect of time of day.

For recordings, flies were immobilized using thin strips of Scotch tape on a pad of pressure-sensitive adhesive (Blu Tack) to maintain the proboscis extended[38]. The body of the fly was grounded using a silver electrode connected to the fly using a conductive gel (Aquasonic 100, Parker). All electrophysiological recordings were made on L2 and L3 sensilla[20].

Combined stimulations/recordings of gustatory sensilla were performed by adapting the tip-recording method[2]. For mechanical stimulation, glass pipettes of 10–15 μm tip diameter were filled with 30 mM tricholine citrate (TCC; Sigma T0252), which blocks responses from the water-sensing gustatory neuron[39]. The pipette was moved over the end of a sensillum (covering ~30–50% of the total length of the shaft). Control steps of 20 μm were delivered using a Piezo micromanipulator (MPC-200, Sutter Instrument) under computer control (Multilink, Sutter instruments). Steps lasted 1 s after which the electrode was returned to the original position. The recording electrode was connected to a taste-specific amplifier (TasteProbe, Syntech)[40], further amplified 10 times, bandpass filtered at 100–3,000 Hz and digitally sampled at 12 kHz (IDAC4, Syntech) under the control of Autospike software (Syntech). Each mechanical stimulation produced a large deflection of the base-line and a train of high-frequency, larger spike-like electrical oscillations was detected lasting ~100 ms (Fig. 4b, region 'ii'), follow by series of small spikes. All of these aspects of electrical responses are severely diminished or lost in *nompC* mutants (Fig. 4c), indicating that they reflect properties of the mechanosensory neuron. For quantifications, we focussed here on the small spikes observed > 100 ms after stimulus onset. Corrected responses for all recordings in the same session were quantified by counting spikes in a 0.5 s window from this time point (Fig. 4b, region iii), subtracting the number of spontaneous spikes in a 0.5 s window before stimulation (Fig. 4b, region i), and doubling the result to obtain spikes s$^{-1}$. The absolute magnitude of responses was variable between sensilla (Fig. 4d); this may be because of variations in the precise direction of displacement of hairs in different animal preparations.

Sucrose stimulation was performed randomly before or after mechanical stimulation using the same type of glass pipettes containing 100 mM sucrose in 30 mM TCC. Responses were quantified by counting the number of spikes in a 1 s window from first contact of the recording electrode with the sensillum[2]. If a sensillum did not show responses to sucrose, the mechanical response was not included in the analysis.

**Statistics.** Sample size was determined based upon preliminary experiments. Data were analysed and plotted using 'R project' (R Foundation for Statistical Computing, Vienna, Austria, 2005; R-project-org) (code available upon request). Data were analysed statistically using different variants of the Wilcoxon test. For comparisons between distributions, the Wilcoxon rank sum test was used. When $P$ value correction for multiple comparisons was required, the Bonferroni method was used. For experiments in Figs 1a, 1e, 2e and Supplementary Fig. 2c, we performed a Wilcoxon signed rank test of the null hypothesis that the median of sampled values differs from zero. Box plots show median plus interquartile range in all figures.

**Data availability.** All relevant data supporting the findings of this study are available from the corresponding author on request.

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

## Acknowledgements

We acknowledge Andrew Lin, Yuh Nung Jan, Martin Göpfert, John Carlson, the Bloomington *Drosophila* Stock Center (NIH P40OD018537) and the Developmental Studies Hybridoma Bank (NICHD of the NIH, University of Iowa) for reagents. We thank Martin Göpfert for advice on mechanosensory physiology, and members of the Benton laboratory for discussions and comments on the manuscript. J.A.S.-A. was supported by a Federation of European Biochemical Societies Long Term Fellowship, an EMBO Long Term Fellowship and a Human Frontier Science Program Long-term Fellowship. F.M.-P. is supported by the French National Research Agency Program DESIRABLE (ANR-12-ALID-0001). Research in R.B.'s laboratory is supported by the University of Lausanne, European Research Council Starting Independent Researcher and Consolidator Grants (205202 and 615094) and a Swiss National Science Foundation Sinergia Grant (CRSII3_136307).

## Author contributions

The authors have made the following declarations about their contributions: Conceived and designed the experiments: J.A.S.-A. and R.B. Performed the experiments: J.A.S.-A. (behaviour, histology and electrophysiology), G.Z. (behaviour). Analysed the data: J.A.S.-A., G.Z., F.M.-P. and R.B. Contributed reagents/materials/analysis tools: J.A.S.-A. and F.M.-P. Wrote the paper: J.A.S.-A. and R.B., with input from all authors.

## Additional information

**Competing financial interests:** The authors declare no competing financial interests.

**How to cite this article**: Sánchez-Alcañiz, J. A. *et al.* A mechanosensory receptor required for food texture detection in *Drosophila*. *Nat. Commun.* **8**, 14192 doi: 10.1038/ncomms14192 (2017).

**Publisher's note**: 

