## [Peer Review File · Nature Communications]

Reviewers' comments:

Reviewer #1 (Remarks to the Author):

This manuscript examines the role of mechanosensory neurons in texture detection in *Drosophila*. The authors show that flies prefer lower agarose concentrations to higher ones and that this preference requires an appetitive substance (sugar or water) and gustatory neurons. They further show that *nompC* mutants, lacking a mechanosensory channel, have defects in texture detection. Overall, this is a nice study that potentially identifies a role for mechanosensation in food discrimination in *Drosophila*. I have some concerns about the assay and the role of *nompC* in texture detection as described below.

1. It is surprising that there is no texture preference in the absence of food. The time course for 0.5 versus 2% agarose (0-90min) in the absence of sugar should be shown to ensure that there is no transient preference. It would also be useful to see the results of a time course of unstarved flies 0.5 versus 2% agarose (plus and minus sucrose).
2. Double labeling with *nompC*-LexA and Gr64f-Gal4, Gr66a-Gal4 and *ppk28*-Gal4 should be performed in order to validate expression of the *nompC* driver in non-gustatory cells.
3. How can the authors exclude a locomotor defect versus a texture defect? *nompC* mutants have trouble walking so the defects in the assay may not be related to texture detection. Silencing the *nompC*-LexA line would be a step towards showing a specific defect.
4. The expression pattern of *nompC*-Gal4 and *nompC*-LexA in the VNC should be shown. If they are both in VNC but only the LexA has a texture defect, this would be consistent with the notion that proboscis chemosensory sensilla are involved.
5. Is the proboscis required for texture discrimination?

Reviewer #2 (Remarks to the Author):

This paper reports the possible role of *nompC*-mediated mechanosensation in texture preference in *Drosophila*. They show behavioral results indicating that *Drosophila* prefer softer agarose in the presence of sugar. Electrophysiological results demonstrate that mechanoreception is impaired or lost in *nompC* mutant flies. *NOMPC* is expressed in labellar mechanoreceptive neurons. The descriptions are interesting and novel as they provide the importance of multimodal sensory integration in feeding behavior. However, the authors fail to present evidence to show how sweet sensory information is modulated by mechanoreception. Thus this manuscript only reports suggestive and preliminary experimental evidence. If they could provide data showing the direct functional interaction, this study could be a novel report.

As major concerns, this manuscript contains many results that are unnecessary and should be deleted.

1. The authors exclude the possibility that visual and auditory information is not need for softness discrimination. These questions are logically possible, but I think these are highly unlikely. It is very hard to imagine that the antennal auditory system can sense agarose softness. These parts (Fig.S2) can be deleted.

2. To know if softness preference is associated with sweet sensation, they test the effect of sugar and water neuron inactivation by TNT. These are OK. They also present data on bitter neuron inactivation. This experiment is apparently unnecessary as they are not using bitter compounds and there is no possibility that bitter sensation is involved in agarose softness detection.

3. Also many of the electrophysiological data can be removed as described below.

4. More detailed behavioral data should be shown for *nompC* flies. These data are essential to claim that this gene is involved in agarose softness sensing. Time course data and data using different agarose concentrations should be shown. At the same time data on other mechanoreceptive mutant strains could be shown.

5. There are several concerns about NOMPC expression pattern as described below (Fig.3).

Fig.1a and 1b each shows results obtained under a same experimental condition. Why PIs using 5 mM sucrose are so much variable?

Fig,1d, I am curious to know why it takes 60 min to turn the preference to 0.5%. The similar tendency is also seen in Fig.S1. Therefore I think it is important to show the time course data for mutants.

Fig.2d, When 1M sucrose solution was spread on 0.5/2% agarose, they think that 2% agarose surface tastes sweeter for flies, but this is just a speculation and this experiment seems to be meaningless.

Fig. 3b

3c, As pointed out above, the *Ga66a* expression pattern is not needed.

3b-d, anti-NOMPC staining reveals small star-like particles. Are these due to non-specific staining? It seems only dendrites are positively stained. Is their previous report showing that NONMC is expressed in dendrites of mechanoreceptor neurons? These points should be explained.

3d, In the *nompC*-LexA/LexAop-GFP and anti-NONMC image, it is expected that one mechanosensory neuron's cell body with an axon and a dendrite will be seen by GFP and anti-NONMC staining will be on the dendrite? However, it is difficult to find such a pattern. The numbers of GFP-positive neurons could be counted to see if that is close to the total numbers of labellar bristles.

Fig.4b, This recording is made by stimulation with 30 mM TCC to inhibit water spikes and two sorts of spikes are observed before a mechanical stimulation. I do not understand why they observe sugar spikes, but this initial recording trace before mechanical stimulation is not necessary.

Fig.4e, They recorded sucrose responses of many strains. These results are not important.

NCOMMS-16-14248-T: RESPONSE TO REVIEWERS

We thank the reviewers for their careful reading and constructive criticisms of our manuscript. Below, we provide responses to each of the raised issues. In addition, we have revised the text to include an expanded Introduction and Discussion. We have also retitled the paper to more clearly highlight the most important findings of our work, i.e., (i) demonstration of texture discrimination behaviour by *Drosophila* when feeding (ii) implication of the mechanosensory channel NOMPC and labellar neurons in this behaviour, and (iii) electrophysiological evidence that these neurons are mechanosensitive, in a NOMPC-dependent manner.

Reviewer #1

This manuscript examines the role of mechanosensory neurons in texture detection in *Drosophila*. The authors show that flies prefer lower agarose concentrations to higher ones and that this preference requires an appetitive substance (sugar or water) and gustatory neurons. They further show that *nompC* mutants, lacking a mechanosensory channel, have defects in texture detection. Overall, this is a nice study that potentially identifies a role for mechanosensation in food discrimination in *Drosophila*. I have some concerns about the assay and the role of *nompC* in texture detection as described below.

1. It is surprising that there is no texture preference in the absence of food. The time course for 0.5 versus 2% agarose (0-90min) in the absence of sugar should be shown to ensure that there is no transient preference. It would also be useful to see the results of a time course of unstarved flies 0.5 versus 2% agarose (plus and minus sucrose).

RESPONSE: We have added the time course of the 0.5% vs 2% agarose without sucrose in Fig. S1b; this shows that no transient strong texture preference is observed in the absence of a sugar stimulus. We have also performed an experiment in which starved and non-starved flies are given a choice between 0.5% and 2% agarose in the presence sucrose (Fig. 2a and Fig. S1a): starved flies show a strong texture preference but, importantly, non-starved flies do not. These results, together with those of Fig. 2c (texture preference with sucrose or sorbitol stimuli) and Fig. 2d (“sweet” neuron inhibition), are all consistent with exhibition of texture preference by flies being dependent upon animals tasting/feeding from an appetitive substrate. This behaviour makes sense in nature: flies can be found upon (non-food) substrates of a variety of textures, but differences in substrates’ textural properties matters most when they are actually feeding.

*2. Double labeling with *nompC-LexA* and *Gr64f-Gal4*, *Gr66a-Gal4* and *ppk28-Gal4* should be performed in order to validate expression of the *nompC* driver in non-gustatory cells.*

RESPONSE: As shown in new Fig. S3a (top), we have performed a series of transgenic double labelling experiments, which confirm that, in the labellum, there is no overlap of cells expressing the *nompC* driver and those expressing gustatory neuron drivers. Moreover, we have visualized the relative projections of these neuron populations in the subesophageal zone (SEZ) (Fig. S3a, bottom). These analyses reveal that the labellar mechanosensory neurons have distinct projections, with no (or, for “sweet” neurons, highly restricted) overlap with gustatory neuron projections.

We note that the lack of substantial overlap of mechanosensory neurons with sweet neurons contrasts with the recent report of (Jeong et al., Nat Comm 2016). This may simply be due to the different drivers used: our *nompC-LexA* driver is restricted to mechanosensory neurons, while the *VT2692-Gal4* and *R41E11-Gal4* enhancer lines used in that study both label, in addition, interneuron soma next to the SEZ. We speculate that some of the overlap with sweet neurons that these authors observed is due to the projections of these interneurons into the SEZ.

3. How can the authors exclude a locomotor defect versus a texture defect? *NompC* mutants have trouble walking so the defects in the assay may not be related to texture detection. Silencing the *nompC-LexA* line would be a step towards showing a specific defect.

RESPONSE: The reviewer highlights a very important aspect of the analysis of *nompC* mutants. Because we observed – as described previously (e.g., Cheng et al., Nature 2010) – that the strongest (putative null) mutants (*nompC¹/nompC³*) show a strong mobility defect, we used this allelic combination only for the histological and electrophysiological analyses (Fig. 3b and Fig. 4), and not in any behavioural experiment.

For the behavioural investigations of the role of *NompC*, we analyzed three hypomorphic allelic combinations, each of which showed defects in texture discrimination, but no overt mobility defects. To confirm our initial qualitative observations of the absence of locomotor phenotypes of these mutants, we have quantified their capacity to explore the plate arena, as shown in Fig. S2b: these experiments confirm that wild-type and *nompC* mutant flies have statistically indistinguishable locomotor/exploratory capacity.

Regarding the suggestion to silence the *nompC-LexA* neurons, please see the response to the comment below.

4. The expression pattern of *nompC-Gal4* and *nompC-LexA* in the VNC should be shown. If they are both in VNC but only the *LexA* has a texture defect, this would be consistent with the notion that proboscis chemosensory sensilla are involved.

RESPONSE: We now provide in Fig. S3b the expression patterns of the *nompC-*

Gal4 and *nompC-LexA* drivers in the VNC, which illustrates that both are indeed expressed in this part of the nervous system, but in different (though possibly partially overlapping) populations of neurons. This is consistent with the modular nature of the regulatory regions of the *nompC* locus as revealed by our original analysis of the brain projections of neuron labeled by these two drivers.

We have also performed neuron inhibition experiments with these drivers (Fig. 3f), which show that the activity of *nompC-Gal4* neurons is dispensable for texture discrimination, while *nompC-LexA* neurons are essential. These data are consistent with a role for labellar neurons in texture discrimination, but do not exclude a contribution from mechanosensors in, for example, the legs (see also response to the comment below).

5. Is the proboscis required for texture discrimination?

RESPONSE: At present we unfortunately do not have genetic reagents that allow us to cleanly dissociate the contribution of NompC-expressing neurons in the labellum from those in the legs for texture discrimination. We attempted to test the role of the proboscis using UV-cured glue to cover the mouthparts; this manipulation led to diminished, but not abolished, texture discrimination (data not shown), suggesting that the proboscis is involved, but that legs also play a role in the decision making process. However, we consider this result tentative, as we observed that glue-treated flies display greatly diminished exploratory behaviour in the arena (possibly due to the long ice-anesthesia they were subjected to immediately prior to the experiment while glue was being applied, or an indirect effect of the spot of glue on the mouthparts).

At this stage, we prefer to be cautious in our interpretations, and not eliminate the possible contribution of both the proboscis and legs to texture discrimination, analogous to the contribution of both labellar and leg sweet sensory neurons in sugar detection (e.g., Thoma *et al.*, Nat Comm 2016). However, we note that the lack of texture discrimination by non-starved flies in the presence of sucrose in the substrate (which would presumably still be activating leg chemosensory neurons) (Fig. 2a and Fig. S1a) argues that flies' expression of texture preference requires contact of the substrate with their labellum as they feed.

Reviewer #2

This paper reports the possible role of *nompC*-mediated mechanosensation in texture preference in *Drosophila*. They show behavioral results indicating that *Drosophila* prefer softer agarose in the presence of sugar. Electrophysiological results demonstrate that mechanoreception is impaired or lost in *nompC* mutant flies. *NOMPC* is expressed in labellar mechanoreceptive neurons. The descriptions are interesting and novel as they provide the importance of multimodal sensory integration in feeding behavior. However, the authors fail to present evidence to show how sweet sensory information is modulated by mechanoreception. Thus this manuscript only reports suggestive and preliminary

experimental evidence. If they could provide data showing the direct functional interaction, this study could be a novel report.

RESPONSE: We are happy that this reviewer finds our data interesting and novel. We highlight that the strengths of our manuscript lie in the behavioural demonstration of texture preference by *Drosophila*, and the characterisation of a mechanosensory channel and neurons necessary for this behaviour, which have not been previously described. Although we provide several lines of evidence that texture preference is revealed only when sugar is present, we consider that this “multisensory integration” may simply reflect that flies only display texture preference when they are actually feeding, using sensory information from their labellar mechanosensors to judge food’s textural properties. We do not examine the converse relationship mentioned by this reviewer, i.e., how texture sensing may impact sweet sensory information processing. We note that this question is the subject of an interesting, complementary paper that was published while we were revising our manuscript (Jeong et al., Nat Comm 2016), and which we now cite in our Discussion. Although that study proposed a direct pre-synaptic regulation of sweet sensing neurons by mechanosensory neurons, we suspect that there will be multiple mechanisms of sensory integration of the chemical and textural qualities of food.

As major concerns, this manuscript contains many results that are unnecessary and should be deleted.

1. The authors exclude the possibility that visual and auditory information is not need for softness discrimination. These questions are logically possible, but I think these are highly unlikely. It is very hard to imagine that the antennal auditory system can sense agarose softness. These parts (Fig.S2) can be deleted.

RESPONSE: We agree that, *a priori*, visual and auditory contributions to texture sensing seem less likely (but certainly not impossible: for example, agarose of different densities is distinguishable by the human eye, and is one way in which we judge food texture remotely). Regarding elimination of the contribution of the NompC auditory neurons, we have now expanded this analysis and incorporated these data into Fig. 3f, to show that while the activity of *nompC-Gal4* auditory neurons is dispensable for texture discrimination, *nompC-LexA* neurons (expressed in the labellum) are essential. Given the multiple different roles of NompC in sensory detection, we consider these data are important in helping to refine knowledge of the neuron population in which NOMPC functions to allow texture discrimination.

2. To know if softness preference is associated with sweet sensation, they test the effect of sugar and water neuron inactivation by TNT. These are OK. They also present data on bitter neuron inactivation. This experiment is apparently unnecessary as they are not using bitter compounds and these are no possibility

that bitter sensation is involved in agarose softness detection.

RESPONSE: We consider the Gr66a>TNT experiment as an important control, showing that the *UAS-TNT* transgene does not itself impact texture preference (in other projects in our lab, we have found that there can be low-level leaky expression of *UAS-TNT* in specific populations of neurons, causing non-specific behavioural phenotypes). In addition, this experiment does argue against the possible contribution of any (unknown) bitter neuron stimulants in the agarose substrate. Nevertheless, we have reordered and rephrased this set of experiments (Fig. 2d and Fig. S1c) to more clearly highlight the Gr66a>TNT genotypes as primarily a control dataset.

3. Also many of the electrophysiological data can be removed as described below.

RESPONSE: see below.

4. More detailed behavioral data should be shown for *nompC* flies. These data are essential to claim that this gene is involved in agarose softness sensing. Time course data and data using different agarose concentrations should be shown. At the same time data on other mechanoreceptive mutant strains could be shown.

RESPONSE: We now provide data on the texture preference of *nompC* mutants at different time points (Fig. S2a), which show that these mutants display diminished preferences throughout the assay. We

Reviewer Figure 1. Texture preference phenotypes of (mechano)sensory ion channel mutants. Preference of wild-type or the indicated mutant strains for 0.5% agarose in a 0.5% vs 2% arena assay in the presence of 5 mM sucrose Wild-type (n=25), *pain*²⁴⁵¹ (n=12), *trp*³⁴³ (n=11), *trp*³⁰² (n=14), *trpA1* (n=12), *pain*²²⁵¹ (n=10), *piezo*^{KO} (n=13).

provide in this document our data on texture preference of various (mechano)sensory ion channels (Reviewer Figure 1). Interestingly, several mutants show diminished texture preference, consistent with the contribution of several pathways that control this behaviour. However, because we have not confirmed these phenotypes with multiple mutant alleles (which is important because, as shown in this Reviewer Figure, *painless* alleles give different phenotypes), we are reluctant to include these preliminary data in the current manuscript. We decided to focus on NompC in the present study because of its strong texture preference defect observed in multiple mutant alleles and because we found it was expressed in the previously uncharacterised labellar sensilla neurons. (We also attempted to test *nanchung* mutants, as these were recently suggested to function in texture preference (Jeong *et al.*, Nat Comm

2016), but we found – as described previously by others e.g., Mendes et al., eLIFE 2013; Akitake et al., Nat Comm 2014 – that these mutants were very sick/uncoordinated, preventing us from confidently assessing their texture preference defects in our assay. Although the neuronal expression pattern of Nanchung was not described in (Jeong *et al.*, Nat Comm 2016), we do not exclude the possibility that NOMPC and Nanchung work together).

5. There are several concerns about NOMPC expression pattern as described below (Fig.3).

RESPONSE: see below.

Fig.1a and 1b each shows results obtained under a same experimental condition. Why PIs using 5 mM sucrose are so much variable?

RESPONSE: We assume the reviewer refers to the data now in new Fig. 2b and 2c. It is an unfortunate fact that *Drosophila* behaviour is inevitably variable, depending upon many known (and unknown) parameters. Although we strove to unify the culture conditions, temperature/humidity of the behaviour room, and arena preparation protocols, we nevertheless found some cases of quantitative variation in the behaviour of flies in similar assays, most often when comparing assay performed at different times of the year (as in this case, when they were performed >9 months apart). However, we note that each set of experiments was performed in parallel with its own set of internal controls to allow us to confidently interpret differences in behaviour within a particular experiment, even if comparison between experiments may be harder.

Fig,1d, I am curious to know why it takes 60 min to turn the preference to 0.5%. The similar tendency is also seen in Fig.S1. Therefore I think it is important to show the time course data for mutants.

RESPONSE: The temporal dynamics of sensory-based decision-making is an interesting question that has only just been started to be investigated (most published gustatory behaviour data report only the preference index at the end of the assay, without regarding to temporal evolution of the phenotype). But it is clear that within the resolution of the assays so far employed, preferences can take time to be revealed. For example, in a two-choice feeding assay *Drosophila* takes at least 10 minutes to showing evidence of decision-making (Ro et al., PLOS ONE 2014). In the texture assay in Fig. 1d, while the behavioral preference only plateaus at around 60 minutes, flies do show a statistically significant preference for 0.5% agarose after 20 minutes.

Concerning the *nompC* phenotype, as mentioned above, we now present the time course data for these mutants (Fig. S2a), which shows that they display diminished preferences throughout the assay.

Fig.2d, When 1M sucrose solution was spread on 0.5/2% agarose, they think that

2% agarose surface tastes sweeter for flies, but this is just a speculation and this experiment seems to be meaningless.

RESPONSE: We performed this experiment as one way to circumvent the possibility that flies prefer softer agarose simply because the sucrose is more available in this less dense substrate (i.e., it tastes sweeter for flies). To test if this was in fact the case, we reasoned that by spreading the sucrose on the top of the agarose and allow it to be absorbed, the concentration of sucrose at the surface would be higher on the denser agarose (because it could diffuse less far in this substrate), which would consequently result in flies preferring the higher density substrate. However, this is not what we observed: flies continue to prefer the softer agarose, suggesting that it is the substrate texture, and not the availability of sucrose, that determines the preference.

This experiment is complemented by the subsequent experiment (Fig. 2f) where we reveal that with homogenous, optogenetic stimulation of sugar neurons (in the absence of external sugar) flies display a texture preference. We feel together these different approaches reinforce each other and would prefer to keep them both in the manuscript.

Fig. 3b 3c, As pointed out above, the Ga66a expression pattern is not needed.

RESPONSE: Given that the mechanosensory neuron in gustatory sensilla has never been characterised in detail molecularly or electrophysiologically, for completeness, we feel it is helpful to show that the NOMPC-stained sensory cilia are distinct from those of bitter sensing Gr66a neurons.

3b-d, anti-NOMPC staining reveals small star-like particles. Are these due to non-specific staining? It seems only dendrites are positively stained. Is their previous report showing that NONMC is expressed in dendrites of mechanoreceptor neurons? These points should be explained.

RESPONSE: The star-like particles are indeed non-specific staining of the antibody in the labellum, as they are still present in *nompC* mutant tissue. NompC protein localizes to the dendrites of mechanosensory neurons and we have not detected labeling of the soma or the axons of these neurons with the antibody (similar to previous reports examining other classes of mechanosensory neuron e.g., Cheng et al., Neuron 2010).

3d, In the *nompC*-LexA/LexAop-GFP and anti-NONMC image, it is expected that one mechanosensory neuron's cell body with an axon and a dendrite will be seen by GFP and anti-NONMC staining will be on the dendrite? However, it is difficult to find such a pattern. The numbers of GFP-positive neurons could be counted to see if that is close to the total numbers of labellar bristles.

RESPONSE: Due to the relatively weak signal of the anti-NOMP staining (and the unavoidable non-specific signal detected with this reagent), it was difficult to

unambiguously identify an anti-NOMPC dendrite for every sensillum. We have however now performed quantitative analysis with the *nompC-LexA>LexAop-CD8:GFP* reporter, which allows more reliable, background-free examination of NompC expression. These flies exhibit GFP signal in the sensory cilia associated with a single neuron in essentially all sensilla of the labella (in rare cases, we were unable to relate a particular sensory dendrite to a cell body). This analysis is consistent with a NOMPC-positive neuron being housed in each sensillum of the labellum.

Fig.4b, This recording is made by stimulation with 30 mM TCC to inhibit water spikes and two sorts of spikes are observed before a mechanical stimulation. I do not understand why they observe sugar spikes, but this initial recording trace before mechanical stimulation is not necessary.

RESPONSE: The presence of spikes prior to mechanical stimulation in the presence of TCC could be due to the spontaneous activity of any of the other three (non-water) gustatory neurons housed in L-type sensilla (we cannot definitively ascribe them to, or distinguish them from, the sweet neuron). The initial trace before the mechanical stimulus is necessary, as it allows comparison of the post- and pre-stimulus neural activity for quantification purposes, and we consider it important to show that prior to the mechanical stimulus the number of spikes is low. This data presentation format is necessarily distinct from that typically used to display gustatory neuron activity, where the traces are shown from the point of stimulus presentation – as for the sucrose-evoked firing in Fig. 4c – because in the tip-recording method, the stimulus is included in the recording pipette.

Fig.4e, They recorded sucrose responses of many strains. These results are not important.

RESPONSE: The sucrose responses measured serve as an important positive control for the mechanosensory responses of sensilla of different *nompC* allelic combinations. We find it is more straightforward to incorporate these quantifications in the same figure as the data for the mechanosensory physiological defects, rather than separating them to a separate figure in the supplementary information.

REVIEWERS' COMMENTS:

Reviewer #1 (Remarks to the Author):

The revised manuscript has addressed my previous concerns and is appropriate for publication.

Reviewer #2 (Remarks to the Author):

The authors respond to all the reviewer's comments and performed additional experiments that look supportive for their data. I recommended to remove several pieces of presented data, while the authors resist to do so. I do not always understand their claims, especially on visual discrimination of softness, but these are not main points and I do not like to comment again. I found the following points that need explanation by the authors. Sorry, this could be pointed out before.

nompC^[f00642] homozygous flies show a mechanoresponse not significantly different from the wild-type (Fig.4d, L2 and L3), though this mutant flies show softness discrimination defect. Two of nompC strains show significantly lower sucrose sensitivities (Fig. 4e). This might be due to genetic background, but it is strange to say nothing about these results.

NCOMMS-16-14248A: RESPONSE TO REVIEWERS

We thank the reviewers for their careful re-reading and constructive criticisms of our manuscript. Below, we provide responses to each of the raised issues.

Reviewer #1

The revised manuscript has addressed my previous concerns and is appropriate for publication.

RESPONSE: We thank the reviewer for her/his comments, which have improved the quality of our manuscript.

Reviewer #2

The authors respond to all the reviewer's comments and performed additional experiments that look supportive for their data. I recommended to remove several pieces of presented data, while the authors resist to do so. I do not always understand their claims, especially on visual discrimination of softness, but these are not main points and I do not like to comment again.

RESPONSE: Regarding the contribution of visual cues, we simply wished to highlight in a sentence in the Results that although agarose of different densities have a different visual appearance (for example, in Figure 1c, the 2% agarose is slightly darker than 0.5% agarose, presumably due to greater absorption of visible light), this property cannot influence flies' choices because the assays were performed under red-light, to which flies are essentially blind.

I found the following points that need explanation by the authors. Sorry, this could be pointed out before.

1.nompC[f00642] homozygous flies show a mechanoresponse not significantly different from the wild-type (Fig.4d, L2 and L3), though this mutant flies show softness discrimination defect.

RESPONSE: The reviewer is correct in her/his observation that this *nompC* mutant strain shows a clear defect in the behavioural assays (Fig. 3a) but a non-statistical difference from controls in the electrophysiology experiments (Fig. 4d). We highlight that this hypomorphic allele – which likely retains low levels of NOMPC function – shows decreased electrophysiological responses compared to wild-type in both of the sensilla classes characterised, but that the relatively high variability in responses in all strains (due in large part to variation in the direction of displacement of the hair) renders these statistically not significant. It is also important to stress that it is hard to directly compare the behavioural and physiological phenotypes, because while the electrophysiological assays sample at the level of individual mechanosensory neurons (for which we have only examined two sensillar classes), the behavioural experiments report on the “pooled” defects across all neurons that depend on NOMPC for mechanosensory responses. We have added a comment on this point in the text (Fig. 4d legend).

2. Two of *nompC* strains show significantly lower sucrose sensitivities (Fig. 4e). This might be due to genetic background, but it is strange to say nothing about these results.

RESPONSE: We have added a comment in the Results regarding the decrease in sucrose sensitivity in certain *nompC* mutant allelic combinations. As the reviewer suggests, we suspect that these differences are due to genetic background, as there is no correlation with the strength of the *nompC* allelic combinations tested.